# Prevalence of HPV Infection and p16^INK4a^ Overexpression in Surgically Treated Laryngeal Squamous Cell Carcinoma

**DOI:** 10.3390/vaccines10020204

**Published:** 2022-01-27

**Authors:** Roberto Gallus, Tarik Gheit, Dana Holzinger, Marco Petrillo, Davide Rizzo, Gianluigi Petrone, Francesco Miccichè, Gian Carlo Mattiucci, Damiano Arciuolo, Giampiero Capobianco, Giovanni Delogu, Vincenzo Valentini, Massimo Tommasino, Francesco Bussu

**Affiliations:** 1Otolaryngology, Mater Olbia Hospital, 07026 Olbia, Italy; roberto.gallus@materolbia.com; 2Infections and Cancer Biology Group, International Agency for Research on Cancer, World Health Organization, 69008 Lyon, France; gheitt@iarc.fr (T.G.); tommasinom@iarc.fr (M.T.); 3German Cancer Research Center (DKFZ), Division of Molecular Diagnostics of Oncogenic Infections, 69120 Heidelberg, Germany; d.holzinger@dkfz-heidelberg.de; 4Gynecologic and Obstetric Unit, Department of Medical, Surgical and Experimental Sciences, University of Sassari, 07100 Sassari, Italy; capobia@uniss.it; 5Otolaryngology Division, Azienda Ospedaliero Universitaria di Sassari, 07100 Sassari, Italy; davide.rizzo@aousassari.it (D.R.); fbussu@uniss.it (F.B.); 6Dipartimento delle Scienze Mediche, Chirurgiche e Sperimentali, Università di Sassari, 07100 Sassari, Italy; 7Department of Women and Child Health and Public Health, Pathology Area, Fondazione Policlinico Universitario A. Gemelli–IRCCS, 00168 Rome, Italy; gianluigipetrone@hotmail.com (G.P.); damiano.arciuolo@policlinicogemelli.it (D.A.); 8Dipartimento di Scienze Radiologiche, Radioterapiche ed Ematologiche, UOC di Radioterapia, Fondazione Policlinico Universitario A. Gemelli IRCCS, 00168 Rome, Italy; francesco.micciche@policlinicogemelli.it (F.M.); giancarlo.mattiucci@policlinicogemelli.it (G.C.M.); vincenzo.valentini@policlinicogemelli.it (V.V.); 9Radiation Oncology Unit, Mater Olbia Hospital, 07026 Olbia, Italy; 10Dipartimento di Scienze Biotecnologiche di Base, Cliniche Intensivologiche e Perioperatorie, Università Cattolica del Sacro Cuore, 00168 Roma, Italy; giovanni.delogu@unicatt.it; 11Mater Olbia Hospital, 07026 Olbia, Italy

**Keywords:** laryngeal neoplasms, human papilloma virus, cyclin-dependent kinase inhibitor p16, biomarkers

## Abstract

**Background:** The mucosal high-risk (HR) human papillomavirus (HPV) is associated with oropharyngeal carcinogenesis. Aims of this study were to evaluate the prevalence of HR-HPV infection in laryngeal squamous cell carcinoma (LSCC) from different subsites, and the clinico-biological meaning of p16 overexpression. **Methods:** Ninety-seven LSCCs submitted to primary surgery (*n* = 75) or to post-irradiation salvage laryngectomy (*n* = 22) were evaluated for HR-HPV DNA and RNA using Luminex-based assays. p16 immunohistochemistry was performed. **Results:** HR-HPV DNA from HPV16 was detected in seven cases (8.75%), without significant differences between supraglottic and glottic lesions. HPV RNA was never detected. p16 overexpression correlated with HR-HPV DNA, but the kappa agreement score was poor. HPV DNA showed no impact on prognosis. p16 overexpression was associated with a better survival (OS, RFS) in primarily operated cases, while an inverse association with OS was observed in the salvage surgery group. **Conclusions:** HR-HPV infection appears to have a marginal role in LSCC independent of the anatomical subsite. p16 expression is deregulated in LSCC independent of HPV but displays a prognostic role in patients submitted to primary surgery. The negative predictive role of p16 overexpression in patients undergoing salvage surgery deserves more investigations for validation and elucidation of its clinical relevance.

## 1. Introduction

Human papillomaviruses (HPVs) are known to infect cutaneous and mucosal epithelia. Some types of HPV from genus alpha, in particular HPV16, have been associated with many human cancers, i.e., cervical, oropharyngeal, anal, vulvar, and penile [1,2] and are collectively called high-risk human papillomavirus (HR HPV) genotypes. The role of HPV in the carcinogenic process in humans is mainly mediated by two oncoproteins, E6 and E7 [3]. A total of 31.1% of cancers attributable to infectious agents, amounting to more than 690,000 cancers diagnosed each year (500,000 of which are cervical cancers) are attributed to HR HPV [4]. 

The role of HPV in laryngeal carcinogenesis, postulated back in the 1980s [5], is less prominent than in other sites and is not currently considered clinically relevant [6,7]. In IARC’s Monograph 100B, the evidence for a role of HPV16 and 18 in LSCC is defined as “non-conclusive” [1]. All the available guidelines and protocols dealing with HPV-driven carcinogenesis in head and neck, such as those from the National Comprehensive Cancer Network, the College of American Pathologists, the Royal College of Pathologists, and Cancer Care Ontario (currently under review), only recommend HPV detection in OPSCC [8]. Rates of HPV infection in LSCCs reported in the literature are extremely variable. IARC’s Monograph 90 [9], which included only PCR-based studies, found a positivity rate ranging from 7% to 59% and prominence of HPV16 (74%). Reported detection rates are influenced by the population tested, the methodology used, and possibly the site of origin of the lesion. In fact, a higher prevalence of HPV in LSCCs of the supraglottic and in particular of the marginal area has been reported [10]. 

However, the mere detection of HPV DNA is not indicative of a cancerogenic role of HPV in a certain malignancy. Only detection methods able to demonstrate transcription of E6 and E7 viral oncogenes, such as mRNA detection essays, can give some evidence of an active role of the virus [11]. Studies using such methods are much less common but are of paramount importance in defining the role of HPV in LSCCs [10,12,13,14,15,16,17,18,19] and indicate a prevalence of HPV-driven carcinogenesis outside the oropharynx probably not exceeding 5% of the SCCs, if not lower. 

p16 is an onco-suppressor that negatively regulates cell cycle progression by inhibiting cyclin-dependent kinases, frequently inactivated in many human malignancies [20,21,22,23,24]. Being a well-known epiphenomenon of the E7 mediated unlocking of the G1-S restriction point through the inhibition and degradation of pRb, the overexpression of the p16 protein has been widely used as a surrogate marker of HPV-driven carcinogenesis in a subset neoplasms known to be HPV-related [1]. The clinical reliability of such a role in head and neck cancer, and oropharyngeal squamous cell carcinoma in particular, is currently a relevant matter of debate [10,25,26], yet p16 overexpression is considered a reliable surrogate marker of HPV-driven carcinogenesis in the new American Joint Committee on Cancer (AJCC) staging system for oropharyngeal squamous cell carcinoma (OPSCC) [27,28]. 

The role of p16 as a standalone prognostic marker, disentangled by its relevance as a surrogate marker for HPV, has been investigated in breast cancer, where its expression seems related to a worse prognosis [29], and in oesophageal squamous cell carcinoma, where among conflicting reports some suggest that it may predict a better overall survival [30]. In laryngeal squamous cell carcinoma (LSCC), the expression pattern of p16 has been investigated, but available data regarding its role as a standalone marker are conflicting. Some reports in the literature connect the expression of p16 with a lack of progression to cancer from precancerous lesions [31], a trend toward better survival in RT-treated patients [17], a longer relapse-free survival (RFS) [14,32], and in a recent paper a significant impact both on disease-specific survival and overall survival in surgically treated patients [33]. However, other studies failed to confirm these findings [26,34,35,36].

The main aim of the present study was to investigate the presence of HPV oncogenes in laryngeal cancers through DNA and mRNA detection assays in the same samples, as well as to investigate the role of p16 as a surrogate marker of HPV infection and/or prognostic marker in LSCC. A secondary aim was to verify the hypothesis of a specific HPV carcinogenic role in the supraglottic/marginal larynx.

## 2. Materials and Methods

Ninety-seven LSCC patients submitted to primary and salvage surgery at Policlinico Gemelli Hospital (Catholic University of the Sacred Heart), Rome, between 1998 and 2013 were included. Data about smoking and alcohol consumption had been collected. All patients had undergone a careful work up, as described elsewhere [37,38,39], all cases had been discussed and followed up by the tumour board and the clinical data archived prospectively in a digital archive (SpiderNet/Speed). Upon retrieval from the archive, all cases were restaged with the VIIth version of the AJCC staging system. All patients provided their informed consent. Exclusion criteria were non-squamous histology, denial of informed consent, or scarcity of stored FFPE material in the histopathology archive. The last criterion led to the exclusion of primary early glottic cancers, most often diagnosed through very small samples (often less than 2 mm^3^).

The present study was approved by the local ethical committee of Fondazione Policlinico Universitario A. Gemelli (Prot Mol.HN sf33976/16 rif. A. 20363/13 id. 204).

### 2.1. Preparation of Paraffin Sections and DNA Extraction

Slides, prepared at the Institute of Microbiology of Policlinico Gemelli according to the HPV-AHEAD protocol, were sent to the laboratories of the IARC Infections and Cancer Biology group at Lyon, France, for HPV DNA analysis. DNA was extracted after an overnight incubation of the paraffin tissue sections in a digestion buffer (10 mM Tris/HCl pH 7.4, 0.5 mg/mL proteinase K, and 0.4% Tween 20).

### 2.2. HPV Type-Specific E7 PCR Bead-Based Multiplex Genotyping

HPV DNA positivity was determined by a type-specific multiplex genotyping (TS-MPG) assay combining multiplex PCR and bead-based Luminex technology (Luminex Corporation, Austin, TX, USA). The test uses type-specific primers targeting the E7 region of 19 different hr-HPV (HPV 16, 18, 26, 31, 33, 35, 39, 45, 51, 52, 53, 56, 58, 59, 66, 68a and b, 70, 73, and 82) and 2 low risk HPV (HPV 6 and 11) genotypes. 

Two primers for the amplification of β-globin were included as a positive control for template DNA quality. After PCR amplification, 10 μL of each reaction mixture was analysed by multiplex HPV genotyping (MPG) using Luminex technology (Luminex Corporation, Austin, TX, USA) as described previously [40]. All HPV DNA-positive FFPE specimens and a randomly selected subgroup of approximately 10% of HPV DNA-negative specimens were further analysed for the presence of HPV E6^*^I mRNA. 

### 2.3. HPV RNA Analysis

Total RNA was purified from three pooled sections of the same tissue block using the Pure Link FFPE Total RNA Isolation Kit (Invitrogen, Carlsbad, CA) as described previously [41]. RT-PCR was carried out using the QuantiTect Virus Kit (Qiagen, Hilden, Germany) in a total volume of 25 μL containing 5 μL of 5xQuantiTect Virus Mastermix, 0.25 μL of 100xQuantiTect Virus RT Mix, 0.4 μM of each oligonucleotide, and 1 μL RNA as described previously [42]. The HPV type-specific E6^*^I mRNA assay developed for 20 HR- or pHR-HPV types was applied for the detection of viral transcripts. The assay amplifies a 65–75 base pair amplicon of HPV and an 81 base pair amplicon of ubiquitin C (ubC) cDNA. 

The HPV RNA analysis was performed at the German Cancer Research Center laboratories in Heidelberg, Germany.

### 2.4. p16 Immunohistochemistry

P16 expression was evaluated by IHC through a monoclonal antibody (clone E6H4, CINtec p16 Histology Kit, Mtm-laboratories, Heidelberg, Germany). FFPE sections were deparaffinized with xylene and rehydrated with alcohol. Antigens were retrieved with a 10 min cycle in epitope retrieval solution (pH 9.0, 95–99 °C). The material was then cooled down at room temperature for 20 min. Endogenous peroxidase was blocked with 5 min immersion in 3% hydrogen peroxide solution. Then a 30 min incubation with the p16 antibody at room temperature was performed, followed by an incubation with a secondary goat anti-mouse for p16. Finally, the sample has been developed with a chromogen substrate solution for 10 min (AEC; Dako, Copenhagen, Denmark) and then stained with haematoxylin (Dako), dehydrated, mounted on slides with a permanent mounting medium and covered. Immunoreactivity was evaluated under microscopic vision. Scoring was performed by an expert pathologist, and the semiquantitative method used included only nuclear staining using the German immunoreactive score of Remmele and Stegner (IRS). An intensity score (absent: 0, weak: 1, moderate: 2, strong: 3) and an extension score (0% = 0, 1–10% = 1, 11–50% = 2, 51– 80% = 3, 81–100% = 4) were attributed to each case. A composite IRS score accounting for both was then calculated by multiplying the sheer numbers (0–12). IHC procedures were performed at the Institute of Histopathology, Policlinico Gemelli, Rome, Italy.

### 2.5. Statistics

Two IRS cut-offs (4 and 6, respectively) were tested with regard to prediction of HPV infection and/or prognosis. The first one was already described in the literature for the evaluation of p16 expression [43]; we also evaluated a higher cut-off, closer to the one most used in the literature [44], to reduce false-positive cases, which are the most undesirable from the perspective of treatment deintensification [27].

Statistical analysis was performed using JMP software, release 7.0.1, from the SAS Institute. The correlation between p16 IHC and HPV status was verified using a χ^2^ test. To assess the reliability of p16 IHC as a diagnostic test for HPV-driven carcinogenesis we computed the Kappa value as previously described [10,25]. Survival curves were always calculated from the time of the first diagnosis. For the univariate analysis, Kaplan–Meyer curves were calculated, and significance was evaluated with the Log-Rank tests.

## 3. Results

### 3.1. Clinicopathological Data

The study includes 97 LSCC patients. For every patient, we analysed just one tumour sample coming from the main surgery, which in 75 cases (77.3%) was the primary treatment (with or without adjuvant therapy), and in 22 cases (22.7%) was performed for salvage after recurrence. Table 1 lists the patients’ characteristics and the clinicopathological features. As expected, alcohol consumption correlated with supraglottic site; no other correlation was detected among behavioural risk factors, staging data, HPV DNA, p16 IHC. No correlation between behavioural risk factors and prognosis was detected. At the last follow-up, 2 patients were alive with disease (2.1%), 11 were dead from other diseases (11.3%), 24 were dead from disease (24.7%), and 60 were alive with no evident disease (61.9%). Along their clinical history, 55 patients had a relapse (56.7%) (39 [40.2%] local, 1 [1%] local and distant, 8 [8.2%] locoregional, 3 [3.1%] regional, and 4 [4.1%] distant).

Seventeen samples resulted negative for ß-globin and were excluded for viral DNA/RNA analyses (Figure 1). 

### 3.2. HPV DNA, RNA and p16 Testing Results

Among the 80 cases tested for HPV DNA, 7 (8.75%) tested positive for HR-HPV16 DNA. No correlation was found between the subsite and HPV DNA positivity (*p* = 0.505). All the HPV DNA-positive cases were further tested for viral RNA but found none of them was positive for HPV RNA, although all seven samples resulted positive for the RNA quality control marker, ubiquitin C (ubC). In 2 out of 97 cases, p16 IHC could not be evaluated for insufficient FFPE tissue. Of the remaining 95 samples, 48.42% were positive for p16 overexpression with an IRS cut-off ≥4 (moderate intensity in at least 11–50% of cells), while only 31.57% were positive with a cut-off ≥6 (moderate intensity in 51–80% of cells or strong intensity in 11–50% of cells).

We evaluated the correlation between p16 IHC and HPV DNA status in the 78 cases where both were available, detecting a significant correlation only when considering as cut-off the composite IRS score ≥ 4 (*p* = 0.007 at Likelihood Ratio test) but not when considering the composite IRS score ≥ 6 (*p* = 0.0733 at Likelihood Ratio test). We then checked the reliability of p16 IHC as a surrogate marker for HPV DNA detection, obtaining a poor agreement (Kappa = 0.14 with IRS score ≥ 4 and Kappa = 0.133 with IRS score ≥ 6), deriving from the high number of false-positive cases (Figure 2).

### 3.3. HPV DNA, RNA, and p16 Impact on Prognosis

Next, we evaluated the impact of HPV DNA detection on prognosis, even if transcriptionally active HPV was demonstrated through mRNA testing in none of the cases. HPV DNA detection was not associated with differences in overall survival (OS) (*p* = 0.64 at Log-Rank) (Figure 3A), relapse-free survival (RFS) (*p* = 0.86 at Log-Rank) (Figure 3D), or disease-specific survival (DSS) (*p* = 0.93 at Log-Rank).

p16 overexpression did not show any predictive value for OS (IRS ≥ 4 cut-off *p* = 0.21 at Log-Rank; IRS ≥ 6 cut-off *p* = 0.97 at Log-Rank) (Figure 3B,C) or DSS (IRS ≥ 4 cut-off *p* = 0.85 at Log-Rank; IRS ≥ 6 cut-off *p* = 0.96 at Log-Rank), but higher p16 expression with the IRS ≥ 4 cut-off was associated with a significantly better RFS (IRS ≥ 4 cut-off *p* = 0.013 at Log-Rank; IRS ≥ 6 cut-off *p* = 0.26 at Log-Rank) (Figure 3E,F).

When primary cases (*n* = 75) and post-irradiation recurrences (*n* = 22) were analysed separately, p16 positivity at IHC, with IRS > 4 as cut-off, was associated with a better OS in primarily operated cases (IRS ≥ 4 cut-off *p* = 0.015 at Log-Rank; IRS ≥ 6 cut-off *p* = 0.5 at Log-Rank), and, surprisingly, with worse overall survival in salvage surgeries after irradiation (IRS ≥ 4 cut-off *p* = 0,038 at Log-Rank test; IRS ≥ 6 cut-off *p* = 0.07 at Log-Rank) (Figure 4).

We calculated RFS from the time of first treatment; so, in the salvage surgery group, where the time of surgery and of sample collection coincided with time to recurrence, this oncological endpoint was not considered. When evaluating RFS in primarily operated cases, p16 overexpression, considering a composite score ≥ 4, showed an association with a better RFS (IRS ≥ 4 cut-off *p* = 0.049 at Log-Rank; IRS ≥ 6 cut-off *p* = 0.26 at Log-Rank) (Figure 5). 

## 4. Discussion

LSCC is the only head and neck site with decreasing survival rates in recent decades [43,45]. Such a trend has been linked to the push towards surgical and non-surgical function/organ preservation [38], in the absence of reliable predictive biomarkers to help in treatment selection. With these premises and considering the well-known prognostic role in OPSCC [14,25,46], it is clear why a potential role of HPV in LSCCs has raised great interest. In the present series, conducted with patients mostly coming from Central and Southern Italy, HPV DNA was detected in about 9% of LSCC, well within the wide range (5.7–25%) reported in the most recent literature [10,12,13,14,18,19,41,47,48,49]. Our data do not lend support to the hypothesis of a prevalent role of HPV in supraglottic SCC [10].

Regardless, in the larynx more than in the oropharynx, where the role of HPV infection is proven, the transcriptional activity and in particular the expression of HPV oncogenes E6 and E7 appears fundamental to hypothesize a carcinogenic role. Data from the literature indicate without doubt that the prevalence of HPV-driven carcinogenesis in the larynx estimated with an mRNA-based approach is much lower [10], ranging from 1.5%, to 7%, with figures well consistent with the present results (0%).

Despite the absence of HR-HPV mRNA in the present series, suggesting that the HPV in the DNA positive cases may have not been transcriptionally active, we tested HPV-DNA in relation to oncological outcomes anyway, finding no statistically significant impact on OS, RFS, and DSS. In the literature, whichever is the method used for HPV testing in LSCC, evidence about a prognostic/predictive role outside the oropharynx is lacking as well [12,14,49,50].

The absence of demonstrated predictive relevance of HPV infection in laryngeal oncology, makes the issue concerning the reliability and standardization of HPV detection methods in LSCC less critical than in OPSCC [27]. However, our data confirm that in LSCC, and in general outside the oropharynx, p16 overexpression is substantially independent of HPV infection [10,14,19,34,49,51] and cannot be used to confirm transcriptional activity of HPV in LSCCs, as seen in some studies [18].

Apart from its potential role as a surrogate marker of HPV infection, p16, encoded by the antioncogene *Ink4a*, is an important cyclin-dependent kinase inhibitor. Loss of p16 expression, mostly an epigenetic suppression mediated by methylation, has been described, well before the demonstration of HPV-driven carcinogenesis, to be a frequent molecular abnormality in HNSCC [18,52,53] with a negative prognostic impact in LSCC [31]. Other studies looking for an impact of p16 on prognosis failed to demonstrate a statistical significance for any of the prognostic indexes analysed, even if often describing a consistent trend [14,36]. In the present work, when considering the whole series, p16 overexpression, determined on the lower cut-off, was significantly associated with a longer RFS, without an impact on the other oncological endpoints.

The present series is not homogeneous as for primary treatment, and not either for time of cancer sample collection, including primary surgery specimens, and previously irradiated cancers, with specimens from salvage surgeries, collected after recurrence. This can be considered a bias, but also an advantage, as we could analyse the two groups separately. In primarily operated cases, p16 expression was strongly associated with longer RFS and OS, and appeared as a prognostic marker, substantially independent of HPV status. On the contrary, in the salvage surgery group, recurrent cancers with p16 overexpression seemed to show a significantly worse survival. The last finding is surprising and, taken together with the first one, potentially extremely interesting from the perspective of a molecular prediction of the response to different treatment modalities and ultimately in guiding treatment selection. Nevertheless, it needs further evaluation, as the salvage surgery group was small (22 patients). Furthermore, in this group, the question of whether p16 expression was the same as at diagnosis or changed between the diagnosis and the recurrence, maybe because of irradiation, remains unanswered.

## 5. Conclusions

Our data, along with those reported by other researchers, do not support a strong role of HPV in supraglottic SCC, evidence further consolidated by the extremely low prevalence in cases studied with an mRNA-based approach. However, when considered as a standalone biomarker, we found p16 overexpression to be significantly associated with a longer RFS. The different impact of p16 on primarily operated and salvaged patients is extremely interesting for its potential role in treatment selection, but it needs further evaluation.

## Figures and Tables

**Figure 1 vaccines-10-00204-f001:**
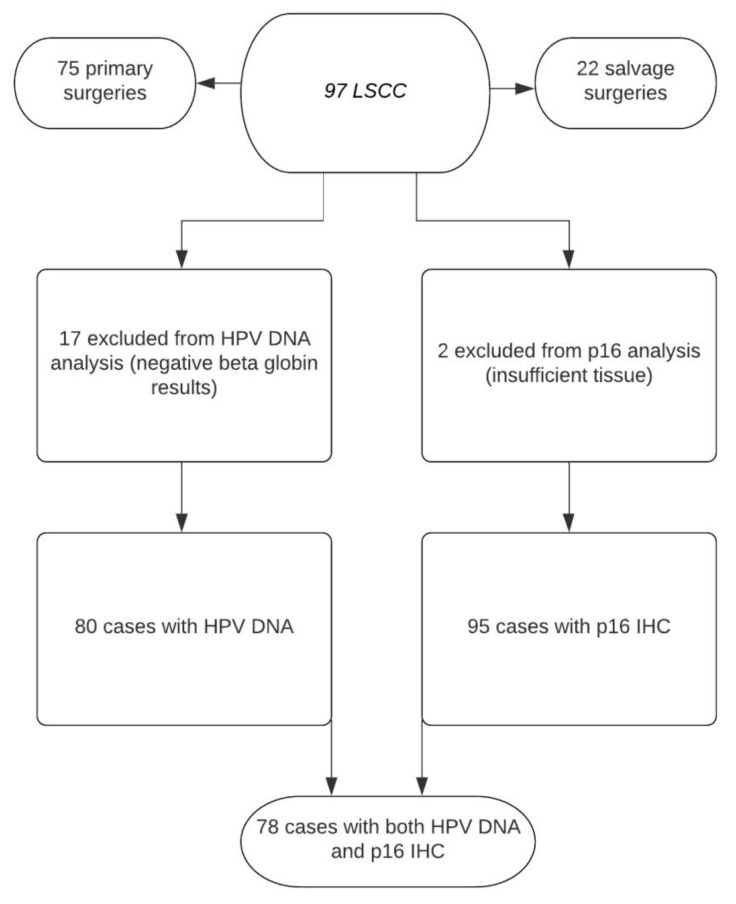
Flowchart describing the availability of data for different patient groups.

**Figure 2 vaccines-10-00204-f002:**
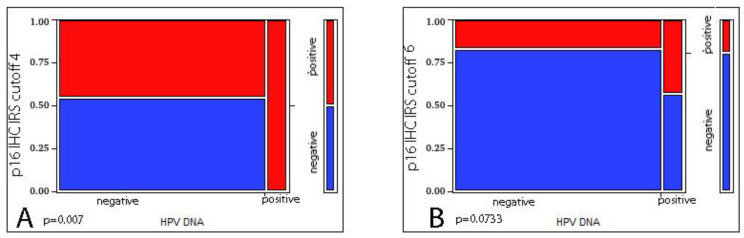
Correlation between p16 IHC and HPV DNA ((**A**) IRS score ≥ 4 *p* = 0.007 at Likelihood Ratio test; (**B**) IRS score ≥ 6 *p* = 0.0733 at Likelihood Ratio test).

**Figure 3 vaccines-10-00204-f003:**
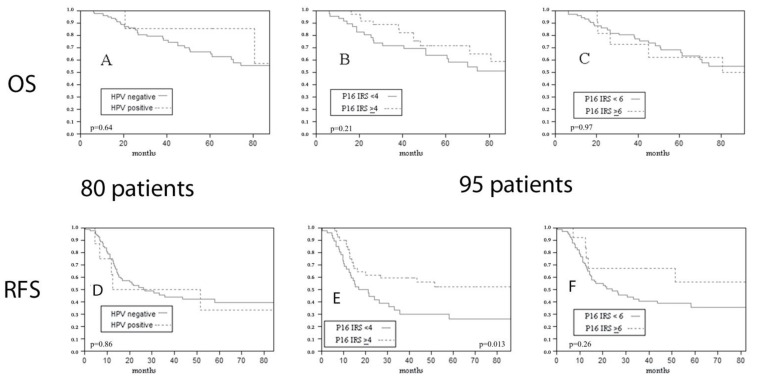
Impact of HPV DNA ((**A**) *p* = 0.64 at Log-Rank) and of p16 expression ((**B**) IRS ≥ 4 cut-off *p* = 0.21 at Log-Rank; (**C**) IRS ≥ 6 cut-off *p* = 0.97 at Log-Rank) on overall survival. Impact of HPV DNA ((**D**) *p* = 0.86 at Log-Rank) and of p16 expression ((**E**) IRS ≥ 4 cut-off *p* = 0.013 at Log-Rank; (**F**) IRS ≥ 6 cut-off *p* = 0.26 at Log-Rank) on relapse free survival of the overall series.

**Figure 4 vaccines-10-00204-f004:**
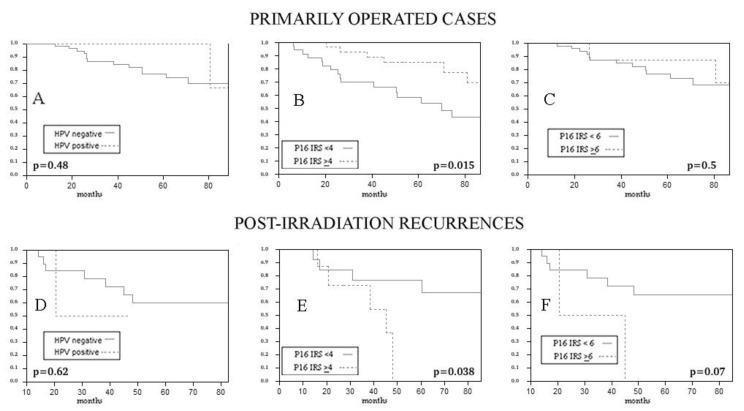
Impact of HPV DNA ((**A**) *p* = 0.48 at Log-Rank; D *p* = 0.62 at Log-Rank) and of p16 expression ((**B**) IRS ≥ 4 cut-off *p* = 0.015 at Log-Rank; (**C**) IRS ≥ 6 cut-off *p* = 0.5 at Log-Rank; (**E**) IRS ≥ 4 cut-off *p* = 0.038 at Log-Rank test; (**F**) IRS ≥ 6 cut-off *p* = 0.07 at Log-Rank), when available, on OS in primarily operated cases (**A**–**C**) and in salvage surgeries (**D**–**F**). p16 overexpression with a IRS score>4 is associated with better survival in primaries (**B**) and to a worse survival in recurrences (**E**).

**Figure 5 vaccines-10-00204-f005:**
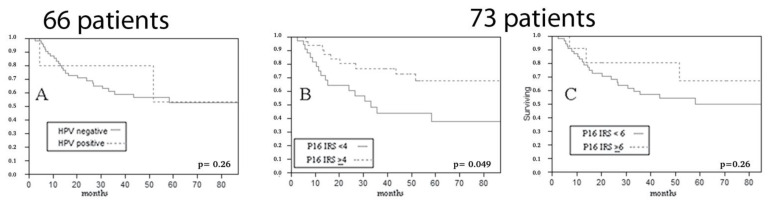
Impact of HPV DNA ((**A**) *p* = 0.26 at Log-Rank) and of p16 expression ((**B**) IRS ≥ 4 cut-off *p* = 0.049 at Log-Rank; (**C**) IRS ≥ 6 cut-off *p* = 0.26 at Log-Rank) on relapse free survival of primarily operated patients (HPV DNA was available in 66 cases, p16 IHC in 73).

**Table 1 vaccines-10-00204-t001:** Personal and clinicopathological data of the patients, stratified in primary and salvage surgeries. Groups were compared by means of Mann-Whitney U test and chi-square test.

	All Cases	First Treatment: Surgery	First Treatment: RT	*p*-Value
**Number of patients**	97	75	22	-
**Age at diagnosis**				
Mean	64.2	64.09	64.73	*p* = 0.529
Range	39.5–88.5	46.35–88.54	39.55–79.25	
**Follow up (months)**				
Mean	55.3	55.57	54.63	*p* = 0.799
Range	1–194	1.07–194.07	12.30–119.63	
**Sex**				
Males	87 (89.69%)	65 (86.7%)	22 (100%)	*p* = 0.071
Females	10 (10.3%)	10 (13.3%)	0	
**Subsite**				
Glottic	53 (54.63%)	37 (49.3%)	16 (72.7%)	*p* = 0.082
Supraglottic	27 (27.83%)	21 (28%)	6 (27.3%)	
Subglottic	1 (1.03%)	1 (1.3%)	0	
Transglottic	16 (16.49%)	16 (21.3%)	0	
**Smoking: current status**				
Current	73 (75%)	65 (86.7%)	8 (36.4%)	*p* < 0.001
Former	19 (20%)	7 (9.3%)	12 (54.5%)	
Never smoker	5 (5%)	3 (4%)	2 (9.1%)	
**Smoking: pack years**				
Mean	29.5	29.2	30.9	*p* = 0.897
Range	0–55	0–55	0–46	
**Drinking: current status**				
Current	76 (78.5%)	65 (86.7%)	11 (50%)	*p* < 0.001
Former	9 (9%)	-	9 (40.9%)	
Never drinker	12 (12.5%)	10 (13.3%)	2 (9.1%)	
**Drinking: alcohol consumption**				
Never	12 (12.5%)	10 (13.3%)	2 (9.1%)	*p* = 0.465
Light	52 (53.5%)	37 (49.3%)	15 (68.2%)	
Moderate	24 (25%)	20 (26.7%)	4 (18.2%)	
Heavy	9 (9%)	8 (10.7%)	1 (4.5%)	
**Staging AJCC at first diagnosis**				
I	17 (17.52%)	8 (10.7%)	9 (40.9%)	*p* = 0.001
II	16 (16.49%)	10 (13.3%)	6 (27.3%)	
III	21 (21.64%)	18 (24%)	3 (13.6%)	
IV	43 (44.32%)	39 (52%)	4 (18.2%)	
**pT at surgery**				
1	16 (16.49%)	8 (10.66%)	8 (36.36%)	*p* < 0.001
2	16 (16.49%)	9 (12%)	7 (31.81%)	
3	27 (27.83%)	23 (30.66%)	4 (18.18%)	
4a	38 (39.17%)	35 (46.66%)	3 (13.63%)	
pN				
N+	28 (28.86%)	27 (36%)	1 (4.54%)	*p* = 0.004
N–	69 (71.13%)	48 (64%)	21 (95.45%)	
**Grading**				
G1	4 (4.12%)	2 (2.7%)	2 (9.1%)	*p* = 0.233
G2	65 (67.01%)	53 (70.7%)	12 (54.5%)	
G3	28 (28.86%)	20 (26.7%)	8 (36.4%)	
**HPV DNA**				
Negative	72/80 (90%)	61/66 (92.42%)	11/14 (78.57%)	*p* = 0.065
HPV11	1 (1.25%)	1 (1.51%)	0	
HPV16	7 (8.75%)	4 (6.06%)	3 (21.42%)	
HPV RNA	-	-	-	
**Staining p16^ink4a^ cut-off ≥** **4**				
Positive	46/95 (48.42%)	37/73 (50.68%)	9/22 (40.9%)	*p* = 0.421
Negative	49/95 (51.57%)	36/73 (49.31%)	13/22 (59.09%)	
**Staining p16^ink4a^ cut-off** **≥** **6**				
Positive	30/95 (31.57%)	24/73 (32.87%)	6/22 (27.27%)	*p* = 0.620
Negative	65/95 (68.42%)	49/73 (67.12%)	16/22 (72.72%)	

## Data Availability

The data presented in this study are available on request from the corresponding author.

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
