# Peer review of "Prevalence of HPV Infection and p16INK4a Overexpression in Surgically Treated Laryngeal Squamous Cell Carcinoma"

_vaccines, 2022, doi:10.3390/vaccines10020204_

Round 1
Reviewer 1 Report
The authors present the clinical data to evaluate the relationship of p16 overexpression with HPV infection in laryngeal squamous cell carcinoma. The clinic data is pretty crucial for better designing laboratory research projects. Although the writing skills can be improved for easier reading, I think it can be published in current styles.
Author Response
Many thanks for the very favorable comments
Reviewer 2 Report
In general terms the presented study has some lack of novelty and it is difficult to follow what is the real interest and/or novelty of the study.
It would be advisable to include other publications to put in context their study:
Effect of HPV Infection on the Occurrence and Development of Laryngeal Cancer: A Review
Dongli Yang, Yong Shi, Yemei Tang, Hongyu Yin, Yujia Guo, Shuxin Wen, Binquan Wang, Changming An, Yongyan Wu, Wei Gao
J Cancer. 2019; 10(19): 4455–4462. Published online 2019 Jul 23.
p16(INK4A) expression in invasive laryngeal cancer
Brenda Y. Hernandez, Mobeen Rahman, Charles F. Lynch, Wendy Cozen, Elizabeth R. Unger, Martin Steinau, Trevor Thompson, Maria Sibug Saber, Sean F. Altekruse, Marc T. Goodman, Amy Powers, Christopher Lyu, Mona Saraiya, The HPV Typing of Cancer Workgroup
Papillomavirus Res. 2016 Dec; 2: 52–55. Published online 2016 Mar 9.
- In the Results section:
“We evaluated the correlation between p16IHC and HPVDNA status in the 78 cases where 165 both were available, detecting a significant correlation only when considering as cut-off 166 the composite IRS score ⪰4 (p=0.007 at Likelihood Ratio test) but not when considering the composite IRS score ⪰6 (p=0.0733 at Likelihood Ratio test). We then checked the reliability of p16IHC as a surrogate marker for HPVDNA detection obtaining a poor agree- 169 ment (Kappa= 0.14) also when using ICS ⪰4 as cut-off”.
The authors need to add the information regarding this part in the manuscript as a Figure.
- In the Figs.2-4, pvalues and relevant statistical information needs to be added to the Figures and Figure Legends
Author Response
Comments and Suggestions for Authors
In general terms the presented study has some lack of novelty and it is difficult to follow what is the real interest and/or novelty of the study.
- While a possible role of HPV in laryngeal squamous cell carcinoma has been extensively studied, studies evaluating comprehensively HPV involvement with different techniques, based on HPVDNA; HPVRNA and p16IHC on the same samples are lacking. This study adds evidence supporting a minimal or no role of alpha HPV genotypes in laryngeal carcinogenesis. Moreover, the potential role of p16 as a standalone prognostic marker deserves some attention, as suggested also by other research groups:
- Sánchez Barrueco A, González Galán F, Villacampa Aubá JM, Díaz Tapia G, Fernández Hernández S, Martín-Arriscado Arroba C, Cenjor Español C, Almodóvar Álvarez C. p16 Influence on Laryngeal Squamous Cell Carcinoma Relapse and Survival. Otolaryngol Head Neck Surg. 2019 Jun;160(6):1042-1047. doi: 10.1177/0194599818821910. Epub 2019 Jan 15. PMID: 30642220.
- Allegra E, Bianco MR, Mignogna C, Caltabiano R, Grasso M, Puzzo L. Role of P16 Expression in the Prognosis of Patients With Laryngeal Cancer: A Single Retrospective Analysis. Cancer Control. 2021 Jan-Dec;28:10732748211033544. doi: 10.1177/10732748211033544. PMID: 34538114; PMCID: PMC8450612.
It would be advisable to include other publications to put in context their study:
Effect of HPV Infection on the Occurrence and Development of Laryngeal Cancer: A Review Dongli Yang, Yong Shi, Yemei Tang, Hongyu Yin, Yujia Guo, Shuxin Wen, Binquan Wang, Changming An, Yongyan Wu, Wei Gao J Cancer. 2019; 10(19): 4455–4462. Published online 2019 Jul 23.
p16(INK4A) expression in invasive laryngeal cancer Brenda Y. Hernandez, Mobeen Rahman, Charles F. Lynch, Wendy Cozen, Elizabeth R. Unger, Martin Steinau, Trevor Thompson, Maria Sibug Saber, Sean F. Altekruse, Marc T. Goodman, Amy Powers, Christopher Lyu, Mona Saraiya, The HPV Typing of Cancer Workgroup Papillomavirus Res. 2016 Dec; 2: 52–55. Published online 2016 Mar 9.
- Thank you for your suggestion, we included both papers and some more to better frame the study.
In the Results section:
“We evaluated the correlation between p16IHC and HPVDNA status in the 78 cases where both were available, detecting a significant correlation only when considering as cut-off the composite IRS score ⪰4 (p=0.007 at Likelihood Ratio test) but not when considering the composite IRS score ⪰6 (p=0.0733 at Likelihood Ratio test). We then checked the reliability of p16IHC as a surrogate marker for HPVDNA detection obtaining a poor agreement (Kappa= 0.14) also when using ICS ⪰4 as cut-off”. The authors need to add the information regarding this part in the manuscript as a Figure.
- Thank you for your suggestion, we added the mosaic plot as figure 2 to better illustrate our results.
In the Figs.2-4, pvalues and relevant statistical information needs to be added to the Figures and Figure Legends
- Thank you for your suggestion, we added p values to figures and figure legends as suggested.
Reviewer 3 Report
The research article submitted by Gallus et al evaluated the prevalence of HR-HPV infection in laryngeal squamous cell carcinoma (LSCC) from different subsites, and the clinico-biological meaning of p16 overexpression. The study looks relevant but the writing part made it weak and unclear. There are few major limitations of the writing that needs to be addressed. The introduction and result section needs to be elaborated and extended. Language correction must be done. Few major and minor comments are listed below.
Major comments
i) The introduction is very thin. Write more on HPV, provide more statistics on HPV prevalence in carcinogenesis, what is HR-HPV (write full form first and then can abbreviate), statistics on p16 detection to consider as marker in other cancers, etc.
ii) Elaborate method 2.1 DNA extraction. Please write reference on Line 94.
iii) Coordinate the result section in two parts under atleast three subheadings 3.1 (For eg: 3.1 Clinicopathological data of the patients), 3.2 , 3.3 ….
Minor comments
i) Correction Line 44: The role of HPV in laryngeal carcinogenesis, postulated back in the eighties was less prominent and is not currently considered clinically relevant.
ii) Correction Line 47: Strike of the word “anyway”
iii) Write full form of AJCC and OPSCC. There are several abbreviations used without mentioning full form anywhere. Please rectify this.
iv) Write small letter for p16 (Line 110)
v) Add a space between HPV DNA/RNA and p16 IHC wherever it is written.
vi) 2.4 – Make the sentence clear starting “Scoring has been performed…” Line 121
vii) Correction Line 157: All the HPV DNA-positive cases were further tested for viral RNA but found none of them was positive for HPV RNA, although all 7 samples resulted positive for the RNA quality control marker, ubiquitin C (ubC).
viii) Correction in Line 195. “done”
ix) Correction Line 199: Strike of the word “anyway”. (I think it is not nice to tell your views with the word anyway for a scientific publication)
x) Is it 1% Line 203
Author Response
Comments and Suggestions for Authors
The research article submitted by Gallus et al evaluated the prevalence of HR-HPV infection in laryngeal squamous cell carcinoma (LSCC) from different subsites, and the clinico-biological meaning of p16 overexpression. The study looks relevant but the writing part made it weak and unclear. There are few major limitations of the writing that needs to be addressed. The introduction and result section needs to be elaborated and extended. Language correction must be done. Few major and minor comments are listed below.
Major comments
- i) The introduction is very thin. Write more on HPV, provide more statistics on HPV prevalence in carcinogenesis, what is HR-HPV (write full form first and then can abbreviate), statistics on p16 detection to consider as marker in other cancers, etc.
- We expanded the introduction as requested.
- ii) Elaborate method 2.1 DNA extraction. Please write reference on Line 94.
- We cited the paper reporting the first description of the procedure.
iii) Coordinate the result section in two parts under atleast three subheadings 3.1 (For eg: 3.1 Clinicopathological data of the patients), 3.2 , 3.3 ….
- We added subheadings as requested.
Minor comments
- i) Correction Line 44: The role of HPV in laryngeal carcinogenesis, postulated back in the eighties was less prominent and is not currently considered clinically relevant.
- Thank you, we edited as suggested.
- ii) Correction Line 47: Strike of the word “anyway”
- Thank you, we edited as suggested.
iii) Write full form of AJCC and OPSCC. There are several abbreviations used without mentioning full form anywhere. Please rectify this.
- We wrote the full form of all the abbreviations used at first appearance.
- iv) Write small letter for p16 (Line 110)
- Edited, thanks.
- v) Add a space between HPV DNA/RNA and p16 IHC wherever it is written.
- Edited, thanks.
- vi) 2.4 – Make the sentence clear starting “Scoring has been performed…” Line 121
- Edited, thanks.
vii) Correction Line 157: All the HPV DNA-positive cases were further tested for viral RNA but found none of them was positive for HPV RNA, although all 7 samples resulted positive for the RNA quality control marker, ubiquitin C (ubC).
- Edited, thanks.
viii) Correction in Line 195. “done”
- Edited, thanks.
- ix) Correction Line 199: Strike of the word “anyway”. (I think it is not nice to tell your views with the word anyway for a scientific publication)
- Edited, thanks.
- x) Is it 1% Line 203
- 5%, the comma should have been a dot, edited.
Reviewer 4 Report
An interesting manuscript for a hot topic involving HPV and head and neck malignancies, the work is well designed and presented. There are just a few comments:
- Abbreviations should be spelled the first time that appear for example in the introduction: AJCC OPSCC.
- “In LSCC, the expression pattern of p16 has been investigated, but available data regarding its role as a standalone marker are conflicting.” Please elaborate more on this statement and support it by relevant bibliography.
- The authors evaluated IRS score cut offs 4 & 6, please provide an explanation for this selection, why not a single cut-off of 5? Preferably support your decision by relevant bibliography.
- Table 1, for the personal and clinicopathological data of the patients, please add one more column and provide simple statistical tests (probably Mann Whitey U tests and Fisher exact) to compare the two populations, thus the interesting reader is aware of possible differences. In my opinion no need to comment on the statistical tests outcomes.
- Figure 1 is confusing, a typical top-down design is easier to understand.
Author Response
Comments and Suggestions for Authors
An interesting manuscript for a hot topic involving HPV and head and neck malignancies, the work is well designed and presented. There are just a few comments:
Abbreviations should be spelled the first time that appear for example in the introduction: AJCC OPSCC.
- Edited, thanks.
“In LSCC, the expression pattern of p16 has been investigated, but available data regarding its role as a standalone marker are conflicting.” Please elaborate more on this statement and support it by relevant bibliography.
- We added the relevant bibliography and expanded as suggested.
The authors evaluated IRS score cut offs 4 & 6, please provide an explanation for this selection, why not a single cut-off of 5? Preferably support your decision by relevant bibliography.
- Thank you for this question, the reason for our choice is briefly explained in the “statistics” section. Usually in head and neck cancer a standard cutoff of >70% cells with moderate to strong staining is applied in a rather arbitrary way as it is simply mutated from the experience with cervical cancer (Begum S, Gillison ML, Ansari-Lari MA, Shah K, Westra WH. Detection of human papillomavirus in cervical lymph nodes: a highly effective strategy for localizing site of tumor origin. Clin Cancer Res. 2003 Dec 15;9(17):6469-75. PMID: 14695150.). This is a choice that has already been a matter of debate (Shelton J, Purgina BM, Cipriani NA, Dupont WD, Plummer D, Lewis JS Jr. p16 immunohistochemistry in oropharyngeal squamous cell carcinoma: a comparison of antibody clones using patient outcomes and high-risk human papillomavirus RNA status. Mod Pathol. 2017 Sep;30(9):1194-1203. doi: 10.1038/modpathol.2017.31. Epub 2017 Jun 16. PMID: 28621317.). In our case, we wanted a more elastic scoring system, also because we wanted to evaluate p16 as a standalone prognostic marker. There is an extremely limited number of papers on IRS applied to p16, and our final selection of cutoffs was meant to include both a cutoff similar to the one usually reported (IRS 6) inclusive of a subset of patients with stronger staining in up to 80% cells, and a second one also reported in the literature (IRS 4, the paper is cited in the text) that includes also all cases with moderate staining in at least 50% of cells. We tried to better clarify this point also in the text.
Table 1, for the personal and clinicopathological data of the patients, please add one more column and provide simple statistical tests (probably Mann Whitey U tests and Fisher exact) to compare the two populations, thus the interesting reader is aware of possible differences. In my opinion no need to comment on the statistical tests outcomes.
- Thank you for the suggestion, we modified the table as requested.
Figure 1 is confusing, a typical top-down design is easier to understand.
- Thank you for your suggestion, we modified figure 1.
This manuscript is a resubmission of an earlier submission. The following is a list of the peer review reports and author responses from that submission.
Round 1
Reviewer 1 Report
Dear Authors, The research design is appropriate. English language and style are fine. Please add information about the Ethic Committee and the given number (materials and methods). The eighth TNM classification should be used in the paper. The work requires punctuation corrections. p value should be written in italics. The quality of figures ought to be improved. To sum up, this work can be accepted after corrections.Reviewer 2 Report
Aims of the submitted study were to evaluate the prevalence of high risk-HPV infection in laryngeal squamous cell carcinoma (LSCC) from different subsites, and the clinico-biological meaning of p16 overexpression. Ninety-seven LSCCs were involved, 75 of them were submitted to primary surgery and 22 to post-irradiation salvage laryngectomy (n=22). High risk HPV DNA and RNA were investigated by Luminex-based assays, while p16 was detected by immunohistochemistry. Authors detected the HPV DNA, but not the RNA. Also p16 overexpression correlated with HPV DNA detection.
p16 overexpression was associated with a better survival (OS, RFS) in primarily operated cases, while inverse association with OS was observed in the salvage surgery group.
The submitted study is focussed, it is important for the field, and generates a hypothesis and a new research question on why recurrent laryngeal cancers with p16 overexpression seem to show a significantly worse survival.